# The Natural History of *CNGB1*-Related Retinopathy: A Longitudinal Phenotypic Analysis

**DOI:** 10.3390/ijms23126785

**Published:** 2022-06-17

**Authors:** Daniel J. Jackson, Adam M. Dubis, Mariya Moosajee

**Affiliations:** 1Institute of Ophthalmology, UCL, London EC1V 9EL, UK; daniel.jackson.21@ucl.ac.uk; 2Global Business School for Health, UCL, London WC1E 6BT, UK; a.dubis@ucl.ac.uk; 3Moorfields Eye Hospital NHS Trust, London EC1V 2PD, UK; 4Great Ormond Street Hospital for Children NHS Trust, London WC1N 3JH, UK; 5The Francis Crick Institute, Imperial College London, London NW1 1AT, UK

**Keywords:** CNGB1, retinitis pigmentosa, natural history

## Abstract

Cyclic nucleotide-gated channel β 1 (CNGB1) encodes a subunit of the rod cyclic nucleotide-gated channel. Pathogenic variants in CNGB1 are responsible for 4% of autosomal recessive retinitis pigmentosa (RP). Several treatment strategies show promise for treating inherited retinal degenerations, however relevant metrics of progression and sensitive clinical trial endpoints are needed to assess therapeutic efficacy. This study reports the natural history of CNGB1-related RP with a longitudinal phenotypic analysis of 33 molecularly-confirmed patients with a mean follow-up period of 4.5 ± 3.9 years (range 0–17). The mean best corrected visual acuity (BCVA) of the right eye was 0.31 ± 0.43 logMAR at baseline and 0.47 ± 0.63 logMAR at the final visit over the study period. The ellipsoid zone (EZ) length was measurable in at least one eye of 23 patients and had a mean rate of constriction of 178 ± 161 µm per year (range 1.0–661 µm), with 57% of patients having a decrease in EZ length of greater than 250 µm in a simulated two-year trial period. Hyperautofluorescent outer ring (hyperAF) area was measurable in 17 patients, with 10 patients not displaying a ring phenotype. The results support previous findings of CNGB1-related RP being a slowly progressive disease with patients maintaining visual acuity. Prospective deep phenotyping studies assessing multimodal retinal imaging and functional measures are now required to determine clinical endpoints to be used in a trial.

## 1. Introduction

CNGB1 (MIM #600724) is located on chromosome 16, with 5657 bp (cDNA) and is formed of 33 exons. It encodes the 240-kDa β-subunit of the cyclic nucleotide-gated ion channel located in rod photoreceptor plasma membranes, consisting of a glutamic acid rich protein (GARP) and channel domain [1]. The CNG channels in rods form heterotetramers consist of 3 α-subunits (CNGA1) and 1 β-subunit (CNGB1), whereas the cone channel is formed by 2 α-subunits (CNGA3) and 2 β-subunits (CNGB3), and both translate light-mediated changes of second-messenger cyclic guanosine monophosphate into voltage signals [2,3,4]. Pathogenic variants in CNGB1 account for up to 4% of autosomal recessive retinitis pigmentosa (RP) cases (RP45 #613767) [5,6,7,8,9]. Patients present with nyctalopia in childhood due to the initial loss of rod photoreceptors, followed by progressive constriction of the visual field. Fundus appearance is typical of RP bone spicule pigmentary deposits and retinal vessel attenuation. Spectral-domain optical coherence tomography (SD-OCT) shows progressive loss of the ellipsoid zone (EZ) and fundus autofluorescence (FAF) demonstrates a hyperautofluorescent (hyperAF) ring around the fovea and areas of hypoautofluorescence corresponding to the atrophic areas and pigment deposition. Despite symptoms developing early in life, visual acuity is usually well-preserved into adulthood, hence patients potentially have a longer therapeutic window for intervention following disease onset compared to other forms of RP [10].

There are 84 known disease-causing CNGB1 variants including 24 missense variants, 21 nonsense, 19 splicing defects, 10 small deletions, 1 small insertion, 1 small insertion–deletion, 7 small duplications, and 1 gross deletion, and this is comprehensively discussed in a recent review [11]. The known variants span the entire gene with no known cluster regions. No genotype-phenotype correlations have been identified to date, however there are only limited reports describing CNGB1-RP phenotypes [5,10,11,12,13,14,15,16,17,18]. Animal modelling of CNGB1-related RP has successfully recapitulated the human disease. In mice, Cngb1-X26 model, generated by excising exon 26, significantly impaired rod function was detected at age 2–3 weeks, followed later by cone dysfunction from six months [19]. Similarly, a canine model has been identified with a spontaneous CNGB1 mutation, c.2387delA;2389_2390insAGCTAC in exon 26, leading to a frameshift and premature stop codon, which closely resembles the Cngb1-X26 mouse and human RP phenotype [20]. Gene augmentation therapy using adeno-associated viral (AAV) vectors has successfully rescued the phenotypes resulting in restoration of structural retinal integrity [12,21]. In the mouse model, a 221 bp long SV40 polyA sequence with a 471 bp mouse rhodopsin promoter was packaged into an AAV8 capsid and injected into the subretinal space of two week old mice [21]. Restoration of rod-driven light responses was observed in the treated mice as well as superior performance in vision-guided behavioural tests. In the canine model, an AAV5 vector was used to deliver canine CNGB1 under control of a human GRK1 promoter. CNGB1 expression was detected at three months and sustained for 23 months following subretinal injection and a sustained improvement in rod-mediated electroretinogram (ERG) was observed [12].

The identification of clinical trial endpoints can be a challenge in subsets of RP with slow disease progression. Visual acuity is commonly not a suitable metric to assess therapeutic efficacy, as has been found with other related studies assessing rod-predominant degeneration [22,23]. Multimodal imaging has been central to assess disease status, with SD-OCT being one of the most valuable methods for quantifying progressive changes involving the EZ [24,25,26,27]. Constriction of hyperAF rings have correlated with disease progression in other RP subtypes [22,28], and have shown to have detectable changes in area over a 2–5 year period in a small CNGB1-related RP cohort of three patients [12]. Central retinal thickness is not a suitable metric due to confounding factors including epiretinal membrane (ERM) and cystoid macular oedema (CMO), prevalent in these patients [11]. This is in line with a previous FDA assessment of age-related macular degeneration (AMD), whereby considerable variability has been previously shown in metrics such as visual acuity, retinal thickness, and visual fields, whereas EZ length and hyperAF ring area may be more useful in measuring photoreceptor degeneration [29].

Longitudinal natural history studies combining structural and functional outcome metrics are required to define prognosis. CNGB1-related RP is a clinical trial priority given the preclinical success of AAV gene replacement. Here we report the natural history of the largest patient cohort of CNGB1-related RP over a 17 year follow-up period. 

## 2. Results

### 2.1. Demographics and Visual Acuity

This study identified 33 patients from 32 families, of which 61% (20/33) were female. The mean age at diagnosis was 40.8 ± 17.1 years (range 10–81 years). The mean follow-up period was 4.5 ± 3.9 years (range 0–17) with a mean number of visits per patient of 5 ± 3.7 (range 1–15 visits). Of these, 21% (7/33) had a family history of RP and 6% (2/33) were from consanguineous families. The cohort consisted of families with diverse ethnic origins: 17 (51.2%) were White British, 10 (30.3%) were Middle Eastern, 2 (6.1%) South Asian, 2 (6.1%) White Other, 1 (3.0%) North African, 1 (3.0%) Black African. The presenting complaint was nyctalopia in 88% (21/24) with symptom onset in childhood for the majority. Of the 33 patients, 18 (54.5%) had homozygous mutations (9 missense, 4 splice site, 3 nonsense, 1 deletion, 1 duplication) with the remaining 15 (45.4%) having compound heterozygous mutations (Appendix A). There were 2 novel variants (c.1936C>T, c.290G>C). Figure 1 details the locations of the mutations, which are distributed across the GARP and channel domains. 

Mean best corrected visual acuity (BCVA) analysis from the right eye of the 33 patients at baseline was 0.29 ± 0.41 logarithm of the minimum angle of resolution (logMAR) (range 0.0–2.2). Twenty-nine patients had at least one follow-up visit and had a mean BCVA of 0.31 ± 0.43 logMAR in their right eye at baseline and 0.47 ± 0.63 logMAR at the most recent (final) visit over a duration of 4.5 ± 3.9 years. Correlation between the change in BCVA at baseline and the most recent follow-up visit demonstrated a trend of worsening BCVA over time (Figure 2A). The change in BCVA of each patient at baseline and most recent visit as a function of age is illustrated in Figure 2B. 

### 2.2. SD-OCT Ellipsoid Zone and Associated Features

SD-OCT images were analysed from 65 eyes of 33 patients. Epiretinal membrane was present in 47.0% (31/65) of the eyes in 17 patients (51.5%) in at least one scan over the mean 4.5 year follow-up period. Cystoid macular oedema was present in 19.7% (13/65) of eyes in 7 patients (21.2%). A lamellar hole was present in one eye, so this eye was excluded from further analysis.

The EZ was measurable in 23 patients from the right eye. EZ length measured at each visit is shown in Figure 3A. Nine were excluded from the SD-OCT analysis as the EZ extended beyond the boundaries of the image; these nine patients had a mean age of 52.5 ± 13.5 years. The mean rate of constriction was 178 µm ± 161 µm per year (range 1.0–661 µm per year). There was no correlation between rate of EZ length constriction with age (*R*^2^ = 0.009 *p* = 0.48) (Figure 3B). Figure 3C represents the correlation of EZ length against age for each patient who had three visits, each with an interval of 10–14 months. This was best described by a linear or exponential decay, but not logarithmic trend (Appendix A). A fitted linear trend shows a detectable decrease in EZ length evident within a clinical trial setting for all the patients represented (*R*^2^ = 0.194, *p* = 0.004). Figure 3D demonstrates EZ length correlation with age for the subtypes of patients with homozygous CNGB1 mutations. From these data, no specific genotype/phenotype correlations can be inferred, possibly due to limited patient numbers and follow-ups coupled with a diverse range of CNGB1 variants.

### 2.3. HyperAF Area 

Seventeen patients had a hyperAF outer ring phenotype that was measurable. There were a diverse range of FAF phenotypes that did not display a characteristic ring pattern in 10 patients (Figure 4). The difference between the FAF phenotypes was not due to more advanced disease with older age, with the ‘ring’ phenotypes having a mean ± SD age of 45.9 ± 14.3 years (range 17–72) and the other phenotypes having a mean age of 54.5 ± 14.7 years (range 28–81). The genotypes between the two groups were also spread across the GARP and channel domain of the CNGB1 gene, with no genotype-phenotype correlations identified. The hyperAF ring area showed a decline with age (Figure 4A) and hyperAF ring constriction rate was not correlated with age (r^2^ = 0.012 *p* = 0.667) with no clear inflection point (Figure 4B). 

## 3. Discussion

We report the findings of the largest retrospective natural history study involving 33 molecularly confirmed patients with CNGB1-related RP. The mean age at diagnosis in our cohort was 40.8 years, yet the majority of patients reported nyctalopia from childhood. This suggests patients may not manifest further debilitating visual symptoms resulting in engagement with ophthalmic services until several decades later, supporting a lengthy window of therapeutic opportunity. Patient records were only pooled from a single tertiary centre and patients may have been assessed at other facilities prior to referral. The patients within our cohort maintained a BCVA of 0.3 logMAR or better after a mean 4.5 ± 3.9 years (range 0–17) follow-up. Thirteen patients had undergone cataract surgery in at least one eye, with a mean age of first eye cataract surgery of 52.7 years (range 37–69) (*n* = 6), however, the age of surgery was unknown in seven patients. The slowly progressive nature and confounding variables, such as cataract surgery, exclude visual acuity from being a reliable clinical endpoint within a trial setting. 

Multimodal retinal imaging using SD-OCT EZ length and FAF hyperAF ring measurements showed most promise as clinical trial endpoints from this study. EZ length exhibited a detectable reduction within a two-year simulated trial period (Figure 2C). There was no clear inflection point to determine if the rate changed with age. ERM prevalence in our study is higher than in previously reported CNGB1 cohorts (34.8% in Nassisi et al. and 20% in Hull et al.), but has been reported in up to 64% from other RP cohorts, precluding the use of central retinal thickness as a reliable outcome measure [10,11,30]. Abnormal hyperAF ring phenotypes on FAF have been reported in 59–68% of RP patients, depending on genetic cause [31,32]. A recent review identified ring phenotypes on quantitative FAF in 79.3% of RP patients with CNGB1 mutations, similar to our cohort of 69.7% [11]. These differing FAF phenotypes did not appear correlated with age or genotype in this study. The use of wider field imaging modalities may increase the number of patients with measurable FAF ring phenotypes. 

Using FAF and SD-OCT as a rapid, non-invasive and easily gradable method to determine clinical endpoints would be highly advantageous. However, the majority of patients reported in this study were over the age of 40 years, therefore, extrapolation of findings to younger cohorts may be limited and reflects the slow progression of disease. Of 33 patients with imaging, nine were excluded from the SD-OCT analysis as their EZ extended beyond the boundaries of the image. Similarly for FAF, six patients were excluded as their hyperAF ring extended beyond the boundary and could not be measured. This will have important implications in identifying outcome metrics, as these patients with milder diseases will likely significantly benefit from therapy. A limitation of this study is the use of measurements from a single grader, and previous repeatability assessments have demonstrated an inter-observer variability in EZ length recordings of within approximately 250 µm [22]. Fifty-seven percent (8/14) of patients had a change in EZ length of greater than 250 µm during a two year trial simulation (Figure 3C), suggesting the change in this metric over this timeframe is greater than inter-observer differences in grading. Deep learning methods to accurately segment retinal layers have shown promise in other cases of RP and could be expanded to CNGB1-retinopathy, potentially overcoming the shortcomings of manual measurements, although these have only been in small datasets to date [33].

Despite collecting longitudinal follow-up of visual outcomes and multimodal retinal imaging, retrospective data from routine clinic visits has limitations in accurately determining important clinical details, such as age of symptom onset, as well as varying imaging protocols and the use of other functional modalities that can vary from visit to visit. Only 13 patients had visual fields measured with a variety of protocols, and nine patients had once only electrodiagnostic testing as part of their clinical care. Full-field electroretinograms (ff-ERG) have shown demonstratable rod dysfunction in all patients, with scotopic responses attenuated in younger and extinguished in older patients from other CNGB1-RP cohorts [11,12]. Abnormal photopic responses have also been reported across different age groups [12]. Symptomatic visual field loss occurs later in life, with confrontational visual field testing showing variable peripheral loss in RP patients with CNGB1 variants [11]. Interestingly, formal kinetic perimetry has detected visual field loss in a 12 year old patient [10]. Retinal sensitivity measures in the form of full field dynamic and static perimetry or microperimetry can complement multimodal retinal imaging in tracking disease progression. Microperimetry has high reproducibility, as well as good interocular correlation, and can accurately detect disease progression in other forms of RP [34,35]. Full-field stimulus testing may feasibly be used to detect measurable endpoints in patients with very impaired visual acuity and visual field, although there are limitations in accessibility and careful patient instruction and examiner training are required to reduce confounding variables [36,37]. No genotype-phenotype correlations for CNGB1-retinopathy have been previously described and this was confirmed within this study. However, recruiting more patients and undertaking a prospective deep phenotyping study may help reveal potential relationships. 

## 4. Materials and Methods

### 4.1. Subjects

Potential subjects were identified from the prospectively consented Moorfields Eye Hospital Inherited Eye Disease Database for structure/function of genetic diseases (Research Ethics Number: 12/LO/0141). Data for these studies are collected as part of standard of care and retrospectively analysed. Patients with molecular genetic confirmation of disease-causing variants in CNGB1 were identified. For all subjects, a full ophthalmic assessment was conducted at each visit as part of their clinical care, including best-corrected visual acuity and retinal imaging. Data was extracted from the electronic medical records of each patient at Moorfields Eye Hospital and supplemented with written records, where appropriate.

### 4.2. Retinal Imaging

Spectral domain optical coherence tomography (SD-OCT) and fundus autofluorescence (FAF) imaging was performed on all patients using the Heidelberg Spectralis (Heidelberg Engineering, Heidelberg, Germany) with Automated Retinal Tracking. The central SD-OCT B scan was identified by a trained observer as having the least residual inner retinal tissue and thickest outer nuclear layer (ONL) presence. FAF imaging was performed using high power blue light autofluorescence at 30 or 55° depending on which best visualised the residual FAF area. EZ length and hyperAF area was measured using the integrated micrometre calliper tool with the inbuilt Heidelberg Eye Explorer software (Heyex, Heidelberg Engineering, Heidelberg, Germany) by a senior trained grader [38]. For analysis, scans whereby the EZ band or hyperAF ring extended beyond the boundaries of the image were excluded.

### 4.3. Statistics 

Given the high degree of human interocular symmetry, the measurements from the right eye were used for analyses. Most measurements occurred at yearly increments; for those that were not, decline was assumed to be linear over the period of time between observations and, therefore, the change was divided by time and thus the rate of change per year was calculated. Pearson correlations were used to assess the relatedness between progression rates and parameters. All statistical analyses were completed using GraphPad Prism (version 8.0.0 GraphPad Software, San Diego, CA, USA).

## 5. Conclusions

We have demonstrated detectable changes on SD-OCT and FAF imaging in patients with CNGB1-related RP. The natural history confirmed a slowly progressive disease whereby central visual acuity is maintained for a lengthy window of time compared with other RP subtypes. EZ length and hyperAF area may be the most sensitive measures of change and have potential as outcome metrics in future trials for subgroups of CNGB1-related RP patients who display the phenotype within measurable parameters. The ability of microperimetry, dynamic perimetry, full-threshold stimulus testing and electrodiagnostic testing to determine disease progression needs to be examined within a prospective deep phenotyping study to provide greater insights into the key functional clinical endpoints that will offer accurate efficacy metrics in assessing future therapeutics.

## Figures and Tables

**Figure 1 ijms-23-06785-f001:**
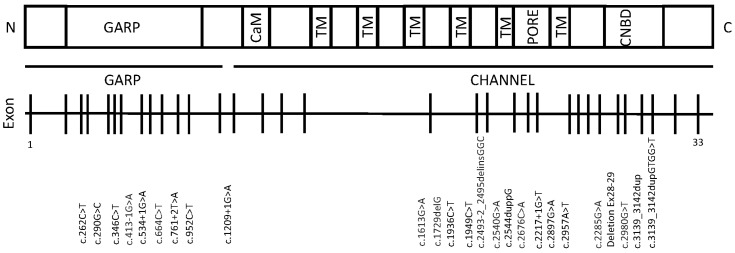
CNGB1 gene and protein schematic with mutations, corresponding to canonical transcript ENST00000251102, NM_001297.5. The protein consists of a glutamic-acid rich protein domain (GARP), a calmodulin-binding domain (CaM), 6 transmembrane (TM) domains, and a cyclic nucleotide binding domain (CNDB).

**Figure 2 ijms-23-06785-f002:**
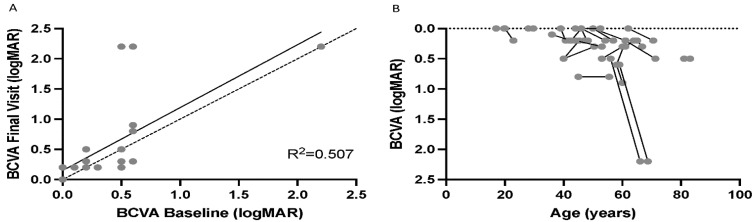
Best corrected visual acuity (BCVA) changes in patients with CNGB1-related RP. (**A**) Pearson correlation between the changes in BCVA from the right eye at baseline to the most recent follow-up. The dashed line represents no change, while the solid correlation line illustrates the change seen in patients. (**B**) demonstrates the change in BCVA of each patient at baseline and most recent visual acuity as a function of age. The majority of patients maintain visual acuity of greater than 0.5 logMAR in at the right eye over the study period.

**Figure 3 ijms-23-06785-f003:**
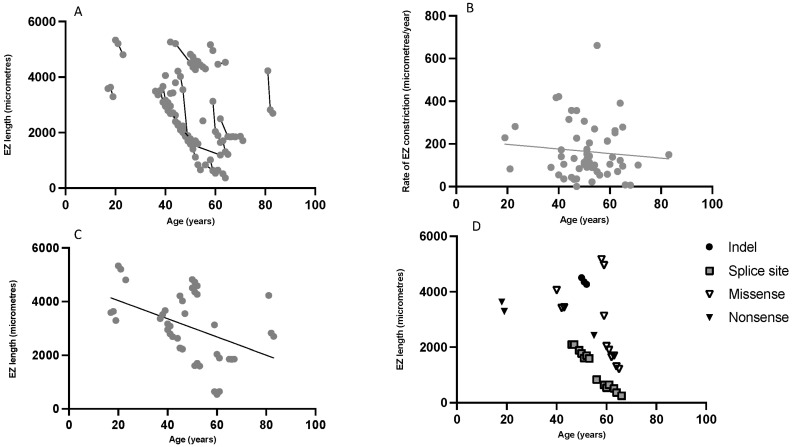
Ellipsoid zone (EZ) length analysis in patients with *CNGB1*-related retinitis pigmentosa. (**A**) Correlation between EZ length and age for each patient at each visit showing a reduction in EZ length. (**B**) EZ constriction rate (change in size divided by time between visits), which appears to be constant with age. (**C**) Correlation between EZ length and age for patients who had three visits, each visit between 10–14 months apart (*n* = 14) with a fitted linear trend that was statistically significant using Pearson correlation (*R*^2^ = 0.194, *p* = 0.004). Fifty seven percent (8/14) had a reduction of EZ length of 250 µm or more during the two year simulated trial period. (**D**) Correlation between EZ length and age for each patient per visit for each homozygous mutation subtype (insertion/deletion ‘Indel’, splice site, missense, and nonsense).

**Figure 4 ijms-23-06785-f004:**
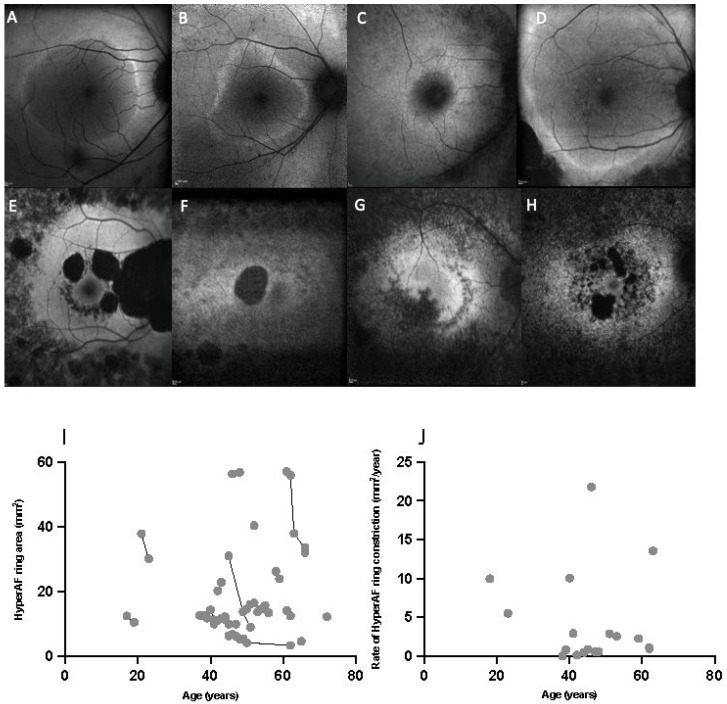
Fundus Autofluorescence (FAF) examples from patients with CNGB1-related retinitis pigmentosa. (**A**–**D**) show a hyperautofluorescent (hyperAF) outer ring phenotype, while (**E**–**H**) do not exhibit a measurable ring pattern. (**I**,**J**) show the quantification of the hyperAF outer ring area on FAF. (**I**) Correlation of hyperAF ring area as a function of age. (**J**) Correlation of rate of constriction of hyperAF area as a function of age, which was not statistically significant (r^2^ = 0.012 *p* = 0.667) using Pearson correlation.

## Data Availability

The summarised data presented in this study are provided in Appendix A. Full datasets are available on request from the corresponding author.

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
