# Peer review of "The Natural History of CNGB1-Related Retinopathy: A Longitudinal Phenotypic Analysis"

_ijms, 2022, doi:10.3390/ijms23126785_

Round 1
Reviewer 1 Report
The ms is well-written exploring an interesting point of view.
I only suggest to better define the statistical approach, in particular in the figure legends. Moreover, the introduction section appears to be too long.
Author Response
We thank you very much for your comments and time taken to review our manuscript.
Comment: I only suggest to better define the statistical approach, in particular in the figure legends
Response: DONE. Statistical calculations and approach has been expanded and added to the legends of Figures 2, 3, 4
Comment: Moreover, the introduction section appears to be too long.
Response: DONE. The introduction has been shortened.
Reviewer 2 Report
The manuscript entitled “The natural history of CNGB1-related retinopathy: a longitudinal phenotypic analysis” deals with the study of the natural history of CNGB1-related Retinitis Pigmentosa with a longitudinal phenotypic analysis of 33 molecularly confirmed patients, with a mean follow-up period 4.5 ± 3.9 years. Mean best corrected visual acuity and Spectral domain optical coherence were measured. The authors evidenced detectable changes on SD-OCT and FAF imaging in patients with CNGB1-related Retinitis Pigmentosa, in Ellipsoid zone length and Hyperautofluorescent area measurements.
The manuscript is well written and easy to understand. The experiments carried out were enough and suitable for the purpose of the manuscript. The references used in the manuscript are recent and adequate. Regarding the novelty of the manuscript, as the authors state, this is the largest retrospective natural history study involving 33 molecularly confirmed patients with CNGB1-related Retinitis Pigmentosa.
In my opinion, the results shown in the present manuscript are interesting for a broader community and deserve to be published. Despite its great potential, the paper comes with a few minor issues which are addressed below:
· Figures are blurry, authors should improve the quality of the images.
· Avoid the use of acronyms and abbreviations in figure legends, as they should be self-explanatory.
· In the figure 4 legend, replace FAF for fundus autofluorescence (FAF).
· Line 81: define AMD.
· Line 111: Define logarithm of the minimum angle of resolution (logMAR)
· Lines 134 and 138: change p and r2 to p and R2
· Line 181: Change LogMAR to logMAR.
· Line 253; Change ONL for outer nuclear layer (ONL)
· The appendix table has very low resolution and appears pixelated.
· Supplementary Figure 1: Spell out EZ length.
Best regards
Author Response
We thank you very much for your comments and time taken to review our manuscript.
Comment: Figures are blurry, authors should improve the quality of the images.
Response: DONE. Higher resolution figures have been submitted in a separate file for final publication
Comment: Avoid the use of acronyms and abbreviations in figure legends, as they should be self-explanatory.
Response: DONE
Comment: Line 81: define AMD.
Response: DONE
Comment: Line 111: Define logarithm of the minimum angle of resolution (logMAR)
Response: DONE
Comment: Lines 134 and 138: change p and r2 to p and R2
Response: DONE
Comment: Line 181: Change LogMAR to logMAR.
Response: DONE
Comment: Line 253; Change ONL for outer nuclear layer (ONL)
Response: DONE
Comment: The appendix table has very low resolution and appears pixelated.
Response: DONE. A higher resolution figure has been submitted in a separate file for final publication.
Comment: Supplementary Figure 1: Spell out EZ length.
Response: DONE
Best regards,
Reviewer 3 Report
Comments CNGB1
The authors report on a large cohort of patients with molecularly-confirmed CNGB1 retinopathy including visual parameters and multimodal imaging features which they suggest as potential endpoints for clinical trials.
Comments are minor and the study is, in general, comprehensive and scientifically sound.
Intro:
Line 31: OMIM uses an asterisk ‘*’ to denote gene names and pound ‘#’ for phenotypes. The 1st use for 600724 should be ‘*.’
Line 49: are there specific cluster regions for mutational hot spots in a certain exon(s)? You discuss exon 26 deletion/variation in dogs/mice. Is this borne out in the known spectrum of human CNGB1 mutation?
Results:
Were any of the causative genetic variants novel?
Line 111: range typo?
Line 132: per year duplicated
Line 157: advanced not advance
Figure 4. Would you not consider case ‘F’ as a measurable ring phenotype? The difference here is the more aggressive outer retinal atrophy outside this. Similar to ‘D’ really but tighter ring.
Would be worthwhile reporting features other than MMI. E.g. what was the phakic refraction here? Does CNGB1 dysfunction drive myopia? How many required cataract surgery and at what age?
Discussion:
Line 200: I suggest highlighting this fact in the results. EZ extended beyond the OCT suggesting widespread preservation of outer retina. What was the mean age of these 9 patients? Similarly, the exclusion of AF could be circumvented using a wider field lens e.g. Spectralis 50 degree, Optos UWV, Clarus.
Line 233: ‘CNGB1-retinopathy’
Author Response
We thank you very much for your comments and time taken to review our manuscript.
Comment: Line 31: OMIM uses an asterisk ‘*’ to denote gene names and pound ‘#’ for phenotypes. The 1st use for 600724 should be ‘*.’
Response: DONE
Comment: Line 49: are there specific cluster regions for mutational hot spots in a certain exon(s)? You discuss exon 26 deletion/variation in dogs/mice. Is this borne out in the known spectrum of human CNGB1 mutation?
Response: Line 53. We have added a sentence confirming there are no known cluster regions. We are unaware of equivalent human mutations as seen in the animal models
Comment: Were any of the causative genetic variants novel?
Response: Line 108. Details of novel variants have been added.
Comment: Line 111: range typo?
Response: DONE - corrected.
Comment: Line 132: per year duplicated
Response: DONE - corrected
Comment: Figure 4. Would you not consider case ‘F’ as a measurable ring phenotype? The difference here is the more aggressive outer retinal atrophy outside this. Similar to ‘D’ really but tighter ring.
Response: Figure 4 image F has been replaced with another FAF image which does not show a ring phenotype.
Comment: Would be worthwhile reporting features other than MMI. E.g. what was the phakic refraction here? Does CNGB1 dysfunction drive myopia? How many required cataract surgery and at what age?
Response: Line 196. We have added the number of patients who required cataract surgery and the age of surgery where known. Due to his being a retrospective study, accurate phakic refractive data was difficult to obtain and therefore has not been reported.
Comment: Line 200: I suggest highlighting this fact in the results. EZ extended beyond the OCT suggesting widespread preservation of outer retina. What was the mean age of these 9 patients? Similarly, the exclusion of AF could be circumvented using a wider field lens e.g. Spectralis 50 degree, Optos UWV, Clarus.
Response: DONE. We have highlighted this point in the results line 134 and have included the mean age of the 9 patients. We have highlighted the use of wider field lens in the discussion line 205.
Comment: Line 233: ‘CNGB1-retinopathy’
Response: DONE